# Analysis of Plant Height Changes of Lodged Maize Using UAV-LiDAR Data

**Longfei Zhou** [1,2,3], **Xiaohe Gu** [1,2,*], **Shu Cheng** [3], **Guijun Yang** [1,2], **Meiyan Shu** [4] and **Qian Sun** [1,2,3]

1    Key Laboratory of Quantitative Remote Sensing in Agriculture of Ministry of Agriculture, Beijing Research Center for Information Technology in Agriculture, Beijing 100097, China; ZLF9510@163.com (L.Z.); yanggj@nercita.org.cn (G.Y.); sunq817@163.com (Q.S.)
2    National Engineering Research Center for Information Technology in Agriculture, Beijing 100097, China
3    College of Geomatics, Shandong University of Science and Technology, Qingdao, Shandong 266590, China; sdchengshu@163.com
4    College of Land Science and Technology, China Agricultural University, Beijing 100193, China; 13383712711@163.com
*    Correspondence: guxh@nercita.org.cn

**Abstract:** Lodging stress seriously affects the yield, quality, and mechanical harvesting of maize, and is a major natural disaster causing maize yield reduction. The aim of this study was to obtain light detection and ranging (LiDAR) data of lodged maize using an unmanned aerial vehicle (UAV) equipped with a RIEGL VUX-1UAV sensor to analyze changes in the vertical structure of maize plants with different degrees of lodging, and thus to use plant height to quantitatively study maize lodging. Based on the UAV-LiDAR data, the height of the maize canopy was retrieved using a canopy height model to determine the height of the lodged maize canopy at different times. The profiles were analyzed to assess changes in maize plant height with different degrees of lodging. The differences in plant height growth of maize with different degrees of lodging were evaluated to determine the plant height recovery ability of maize with different degrees of lodging. Furthermore, the correlation between plant heights measured on the ground and LiDAR-estimated plant heights was used to verify the accuracy of plant height estimation. The results show that UAV-LiDAR data can be used to achieve maize canopy height estimation, with plant height estimation accuracy parameters of $R^2 = 0.964$, RMSE = 0.127, and nRMSE = 7.449%. Thus, it can reflect changes of plant height of lodging maize and the recovery ability of plant height of different lodging types. Plant height can be used to quantitatively evaluate the lodging degree of maize. Studies have shown that the use of UAV-LiDAR data can effectively estimate plant heights and confirm the feasibility of LiDAR data in crop lodging monitoring.

**Keywords:** UAV-LiDAR; lodging maize; plant height; crop height model; recovery ability

## 1. Introduction

According to a report from the National Bureau of Statistics, China's maize planting area in 2019 was 41.284 million hectares, with a total output of 260.77 million tons, which makes maize the largest food crop in China [1]. In recent years, the frequent occurrence of extreme weather events caused by global warming has increased the probability of maize lodging. The maize growing season is from July to September every year. During this period, severe weather events such as strong winds and rainstorms occur frequently, and are the main cause of maize lodging [2,3]. Maize lodging seriously affects the normal progress of plant photosynthesis and nutrient transport. At the same time, it can

easily lead to a variety of diseases and insect pests, resulting in a decrease in the number of grains per ear and 1000-grain weight, and thus seriously affecting maize grain yield, quality, and mechanical harvesting ability [4–6]. Lodging occurs in different periods and degrees, and the impact on output differs. In mild cases, the output is reduced by about 12%, and in severe cases, it is reduced by more than 75%, or even 100% with no harvest [7,8]. Lodging, as one of the major disaster stresses in maize production, is a serious threat to food security. Understanding the scope and degree of lodging disasters over time is of great significance for the agricultural sector to guide agricultural production and disaster assessment in maize lodging [9,10].

At present, the main methods of monitoring crop lodging are field-based and remote sensing methods [11]. In the field-based methods, relevant lodging information is recorded through field investigation of lodged areas by investigators, which is time-consuming and inefficient. In the remote sensing method, lodging information is extracted according to the specific values of various characteristic remote sensing-derived variables (spectrum, hue, texture, etc.) of lodging and non-lodged plots in remote sensing images [12–14].

Most crop lodging studies are based on field crop spectra or the use of multi-spectral, hyperspectral, and other optical sensors to perform data analysis. Field crop spectra can be collected using an ASD FieldSpec Pro spectrometer. Zhang et al. [15] collected ASD hyperspectral data on the ground, used a wavelet transform to process and analyze the data, established a model to predict grain quality parameters, and evaluated the differences of grain quality between non-lodging and lodging maize. Shu et al. [16] used lodged winter wheat at the filling stage as the research object to classify the lodged degree of lodged wheat (severe: lodged angle ≤ 15°; moderate: 10° < lodged angle < 45°; mild: 45° < lodged angle < 70°; non-lodging > 70°), and explored the changes in canopy structural characteristics of winter wheat and its canopy spectral response under different lodged intensities. Although ground-based ASD hyperspectral data have high spectral resolution and can reflect the subtle differences in the spectra of different lodging degrees, such data can only be measured at a fixed point, are not suitable for large-scale lodging monitoring, and can only be used as a priori knowledge of crop lodging stress in remote sensing monitoring. A variety of sensor images obtained through remote sensing technology have wide coverage and can be used for large-scale crop lodging research. Han et al. [17] used an unmanned aerial vehicle (UAV) to obtain multi-spectral and digital images, extracted a variety of characteristic factors related to lodging (such as texture, canopy structure, spectral features, etc.), and used the Nomogram diagram method to analyze the probability of maize lodging. The results are suitable for extracting and analyzing phenotypic traits of large-scale breeding and agronomic experiments. Singh et al. [18] used validated digital lodging measurements along with association and genomic prediction analyses to provide evidence in support of a polygenic genetic architecture of wheat lodging. Liu et al. [19] studied the canopy spectra of wheat with different lodging angles and found that larger lodging angles were associated with greater canopy spectral reflectance. Based on this finding, the degree of wheat lodging was successfully monitored. Liu et al. [20] used a UAV to obtain visible and thermal infrared images of lodging rice, and found that the color and texture features and canopy temperatures of lodging and non-lodging rice plants were different, and fused these features to establish a comprehensive identification model of rice lodging using particle swarm optimization and a support vector machine algorithm, which significantly improved the accuracy of rice lodging recognition.

However, severe weather such as severe storms and rains is the main cause of lodging [21,22]. Affected by severe weather, the timely acquisition of optical remote sensing data in the lodged area cannot be guaranteed, affecting the monitoring of lodged disasters. At the same time, interference by factors such as soil conditions, water or nutritional stress, pests, and diseases will also cause differences in the optical remote sensing data reflection spectrum, which will have a certain impact on the actual change of the canopy spectrum caused by lodging [23,24]. Therefore, the use of optical data to monitor crop lodging has some limitations. Synthetic aperture radar (SAR) data are not affected by bad weather, can provide continuous time series data, and are very sensitive to structural changes. In theory, using

SAR data to monitor lodging has advantages [25]. Yang et al. [26] successfully distinguished between lodging and non-lodging wheat in the farmland of Shangkuli, Inner Mongolia, China, using continuous time-series Radarsat-2 full-polarization image data covering the entire wheat growth period. Given that SAR data are sensitive to structural changes, Han et al. [21] used the plant height ratio before and after lodging as an evaluation index of the degree of lodging to realize the monitoring of the maize lodging disaster level at the regional scale of Sentinel-1A radar image. Shu [22] and Chauhan [27] respectively used SAR data to estimate crop lodging angles and quantitatively assess the degree of the lodging disaster. However, at present, related research on the application of SAR data in crop lodging monitoring are limited to fully polarized data, and has been oriented toward the plot scale or pixel scale [24,25,28].

After lodging, the morphology of a maize plant can change drastically, and maize plants change from upright to slight leaning to flat horizontal on the ground. The more serious the lodging, the lower the plant height. Therefore, plant height can be regarded as an important index to characterize the degree of crop lodging. Traditional manual measurement of plant height is too time-consuming and labor-intensive, and is subject to human error; therefore, it cannot meet the needs of agricultural production. The common method of crop height monitoring by remote sensing is to generate three-dimensional (3D) point clouds from digital images of a UAV [29,30]. Although the cost of UAV digital images is low, the accuracy and quality of the acquired point clouds are easily affected by weather conditions [31,32]. Compared with optical sensors, light detection and ranging (LiDAR) is an active remote sensing tool that can obtain the three-dimensional coordinate information of the target, and can provide vegetation height and vertical structure information [33,34]. In addition, it is not affected by light conditions [35]. Compared with that obtained from digital images, plant height information obtained by LiDAR is more accurate [36]. At present, LiDAR has been applied to plant height monitoring of different crops. For example, Crommelinck et al. [37] used a terrestrial laser scanner (TLS) to obtain multi-phase maize plant height data, and used a crop height model to extract maize plant heights, which provided a point cloud processing method for real-time monitoring of maize plant height. Jimenez et al. [38] used LiDAR to measure wheat plant height, ground coverage, and aboveground biomass. Of these parameters, the estimation results for wheat canopy height showed $R^2 = 0.99$ and RMSE = 0.017 m, which indicate higher plant height estimation accuracy. In addition, rice [39], tomatoes [40], cotton [41], and other crops have been investigated in this manner. These studies achieved satisfactory results in LiDAR monitoring of plant height.

To date, few applications have been presented for monitoring lodging using LiDAR data. In this study, UAV-LiDAR was used to obtain lodging maize point cloud data at different growth stages, and the acquired LiDAR point cloud data were processed and analyzed to obtain the height of lodged maize at a single growth stage. The objectives of this study were: (1) to analyze the changes in plant height of lodging maize and evaluate the plant height recovery for different lodging degrees, and (2) to use LiDAR point cloud data to estimate the height of lodged maize, and (3) to evaluate the application potential of LiDAR point cloud data in monitoring the height of lodged maize.

## 2. Materials and Methods

### 2.1. Study Area

This study was carried out at the National Experimental Station for Precision Agriculture, which is in the Changping district of Beijing, China (40°10.6' N, 116°26.3' E, Figure 1). This site has a mean annual rainfall of 508 mm and a mean annual temperature of 13 °C [42]. The regional terrain is high in the northwest and low in the southeast, and mainly consists of cinnamon soil and tidal soil. The Jinghua No. 8 maize variety was selected as the test material in the experiment; the growth period of this variety is from July to October each year. At the time of sowing, the soil was loam, the ridge spacing was 70 cm, the plant spacing was 30 cm, the sowing density was about 48,000 plants/ha, the base fertilizer was pure nitrogen (180 kg/ha), and there was no topdressing.

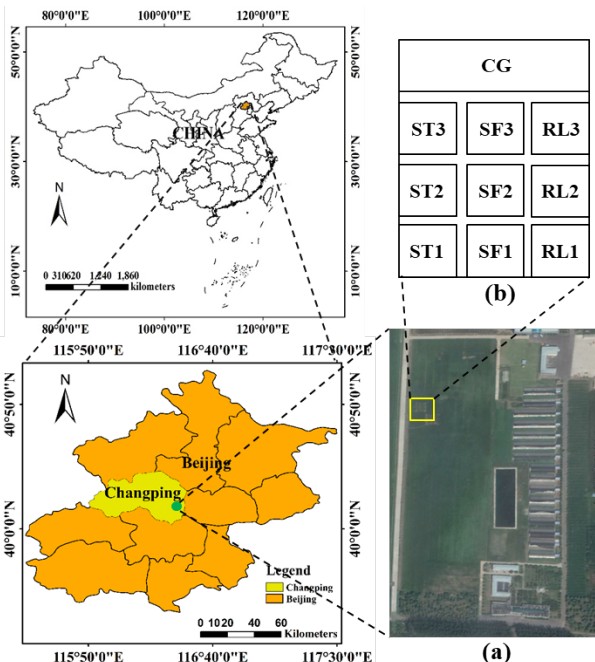

**Figure 1.** Study area location and experimental design. (**a**) Experimental area location. (**b**) Experimental design. Note: CG: control group; ST: stem tilt; SF: stem folding; RL: root lodging.

## 2.2. Experimental Design

The maize lodging experiment started in 2018, and uniform field management was adopted for the maize in the experimental area during the growing period. During the male tasseling period (August 28), an artificial lodging experiment was carried out. Lodging is primarily of two kinds: root and stem lodging. The degree of lodging is from severe to mild, the order was as follows: root lodging (RL)—the main root is broken, the fibrous root on one side is not broken, and the entire anchorage system was destroyed; stem folding (SF)—severe stem lodging, the plant is folded at 20 cm above the ground, but not disconnected; stem tilt (ST)—mild stem lodging, the plant is tilted as a whole, and the angle between the plant and the ground is 45°. The type of lodging is shown in Figure 2. In the east–west direction, there were three different types of lodging treatments, and in the north–south direction, three repeated experiments were conducted for each lodging type; the control group (CG) was located north of the lodging area. The size of the experimental area was 20 m × 25 m. A total of 12 plots were investigated. (Among them, the normal maize plot is regarded as a whole when obtaining plant height data, as shown in Figure 1(a), each plot area was 5 m × 5 m, and protection lines were set up between adjacent experimental plots. An experimental design diagram is shown in Figure 1. Field irrigation treatment was carried out before lodging. The lodging direction of the plants was consistent with the wind direction of the local summer storm weather. In each plot, 5 plants were randomly selected, and the height of the canopy was measured with a telescopic leveling ruler, and 50 samples were finally obtained.

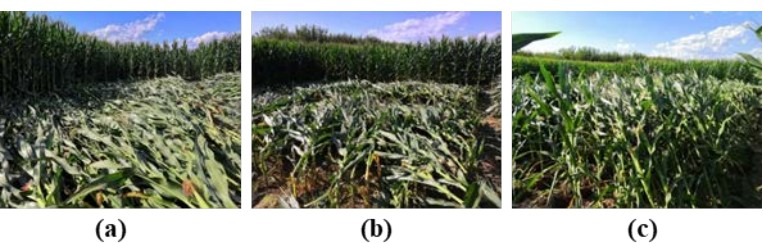

**Figure 2.** Diagram of lodging. (**a**) RL: root lodging. (**b**) SF: stem folding. (**c**) ST: stem tilt.

## 2.3. Data Acquisition

### 2.3.1. Flight Route Design

The UAV-LiDAR data were collected on August 28 (tasseling stage) and September 14 (filling stage), 2018. Data were collected before and after artificial lodging on August 28; the first UAV flight obtained the maize LiDAR data without lodging treatment, and the second flight obtained the maize LiDAR data after artificial lodging. On September 14, the measured plant height data and LiDAR data of the lodging maize were collected. The plant height of maize in the lodging plot was measured manually with a telescopic leveling ruler, and the plant height was measured at five sampling locations selected randomly in each plot. The same sensor parameters and flight path were set for flights on the same day. The drone takes a cross flight, with six routes for each flight (Figure 3). Figure 3 show the route design diagrams for the tasseling and filling stages, respectively, including the distance between the routes, and the location relations between the experimental area and each route. The distance between adjacent routes on August 28 (Figure 3a) was 25 meters, and on September 14 (Figure 3c) the distance between adjacent routes was 15 meters. The point cloud densities (point cloud number in the experimental area/(length × width)) of different routes are shown in Table 1. The spot diameter was 0.0075 m, and the average ground point spacing was 0.0239 m [43].

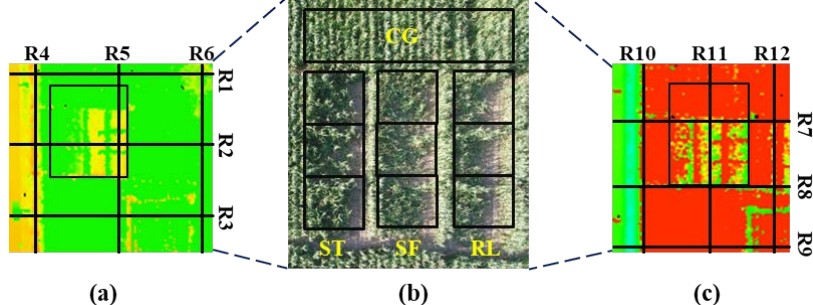

**Figure 3.** Description of the experimental design. (**a**) The experimental area and each flight route on August 28. (**b**) Experimental area. (**c**) The experimental area and each flight route on September 14. Note: R1 to R12 are the flight routes of the UAV-LiDAR. CG: control group; ST: stem tilt; SF: stem folding; RL: root lodging.

**Table 1.** LiDAR point cloud densities of the different routes.

| Direction | Route | Point Cloud Density (pts/m$^2$) | Direction | Route | Point Cloud Density (pts/m$^2$) |
|---|---|---|---|---|---|
| EW | R1 | 112 | NS | R4 | 228 |
| | R2 | 280 | | R5 | 496 |
| | R3 | 529 | | R6 | 417 |
| | R7 | 570 | | R10 | 321 |
| | R8 | 285 | | R11 | 466 |
| | R9 | 200 | | R12 | 366 |

Note: EW indicates the east–west direction, NS indicates the north–south direction, and R1 to R12 are the flight routes of the UAV-LiDAR.

### 2.3.2. Sensor Parameter Settings

The laser scanner mounted on the drone was a RIEGL VUX-1UAV sensor (Figure 4). The related performance parameter settings are shown in Table 2 [44,45]. The UAV had a flight height of 15 m and a flight speed of 3m·s$^{-1}$. The RIEGL VUX-1UAV sensor obtained data by scanning with a rotating mirror, the laser pulse emission frequency was 550 kHz, and the data format was LAS. The flight parameter settings were the same for both flights.

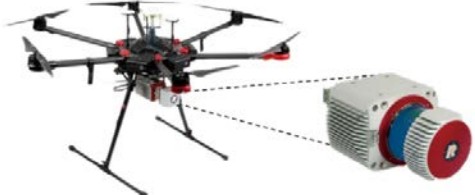

**Figure 4.** Schematic diagram of the RIEGLVUX-1UAV sensor.

**Table 2.** Sensor related parameters.

| Parameter | Value |
|---|---|
| Wavelength (nm) | 1550 |
| Flying speed (m·s$^{-1}$) | 3 |
| Flying height (m) | 15 |
| Area coverage routes (lines) | 6 |
| Scan overlap rate (%) | 40 |
| Pulse frequency (kHz) | 550 |
| Beam divergence angle (mrad) | 0.5 |

### 2.4. Data Processing

Point cloud data processing mainly includes two parts: one is the trajectory data solution, including the original laser data processing and point cloud data preprocessing; the other is point cloud data analysis [46]. The processing steps include: trajectory calculation, raw laser data processing, point cloud and trajectory data matching, and point cloud data processing and analysis. The process flow is shown in Figure 5. First, the POSPac software (version 7.2, Applanix., Richmond Hill, Ontario, Canada) was used to perform precise orbit calculations on GPS base station data and the Position and Orientation System (POS) data. Then, the RiPROCESS software (version 1.7.2, RIEGL Co., Horn, Austria) was used to process the original laser data, including the waveform calculation, coordinate system, 3D point cloud data visualization, and output point cloud data in the LAS standard format after processing. Finally, the LiDAR360 software (version 3.0, GreenValley Co., Beijing, China) was used to analyze the point cloud data, which mainly included point cloud denoising, classification, filtering, and other steps.

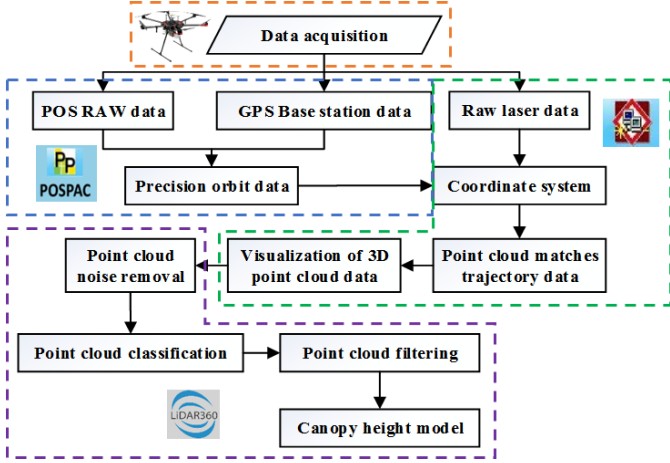

**Figure 5.** Point cloud data processing process.

### 2.5. Canopy Height Model

The canopy height model (CHM) was used to obtain the canopy height of lodged maize. The CHM is an expression of the height of a surface object and reflects the continuous surface distribution of

the object in the horizontal direction, as well as the three-dimensional structure and height change in the vertical direction. To determine the maize canopy height of the lodging plots, a CHM was constructed used in the LiDAR360 software, a triangulated irregular network (TIN) was used for spatial interpolation, and the pixel resolution was set to 0.05 m [47,48]. First, the ground point cloud data were extracted to obtain a digital elevation model (DEM). For the DEM, digital terrain simulation is realized based on limited terrain elevation data. It is a solid ground model that represents ground elevation in the form of an ordered numerical array, and is a branch of the digital terrain model. Secondly, the digital terrain model (DSM) is established according the canopy point cloud data. The DSM is based on the DEM and further covers the elevation information of other surfaces in addition to the ground. For example, it includes the height of buildings, bridges, and trees. In contrast, the DEM contains only the elevation information of the terrain, not other surface information. The relationship between the two is shown in Figure 6. The CHM can be obtained by subtracting the DEM from the DSM. The calculation formula is shown in Equation (1) [49]:

$$CHM = DSM - DEM \tag{1}$$

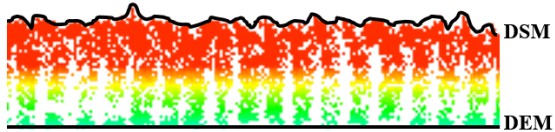

**Figure 6.** Relationship between the DEM and DSM. Note: DSM: digital surface model; DEM: digital elevation model.

## 2.6. Evaluation of Accuracy

In this study, the maize plant height estimated by LiDAR and the measured plant height were verified. The coefficient of determination ($R^2$), root mean square error (RMSE), and normalized root mean square error (nRMSE) were calculated to evaluate the accuracy of LiDAR for estimating plant height [50]. The $R^2$ value was used to evaluate the coincidence between the estimated values and the measured value. The RMSE was used to measure the deviation between the estimated values and the measured values. The nRMSE represents the degree of difference between the estimated values and the measured values (nRMSE <10% indicates no difference, 10%≤ nRMSE <20% denotes a small difference, 20%≤ nRMSE <30% is moderate, and nRMSE ≥30% represents a large difference) [51]. Among them, a larger $R^2$ value indicates better data fit, and smaller RMSE and nRMSE values indicate higher estimation accuracy. The calculation formulas of $R^2$, RMSE, and nRMSE are shown in formulas (2), (3), and (4):

$$R^2 = 1 - \frac{\sum_{i=1}^{n} (y_i - \bar{x})^2}{\sum_{i=1}^{n} (x_i - \bar{x})^2}, \tag{2}$$

$$RMSE = \sqrt{\frac{\sum_{i=1}^{n} (y_i - x_i)^2}{n}}, \tag{3}$$

$$nRMSE = \frac{RMSE}{\bar{x}} \tag{4}$$

where $x_i$ and $\bar{x}$ represents the measured value and the average of the measured values, respectively, $y_i$ represents the estimated value, and n represents the number of samples.

## 3. Results and Analysis

### 3.1. Profile Analysis of the Study Area

The most apparent effect of maize lodging was that the plant heights changed greatly, and the point cloud data directly reflect these changes in plant height. The point cloud data processed on August 28 and September 14 were analyzed in the LiDAR360 software, and the results are shown in Figure 7. The three profiles were selected at the same location, and the profile buffer was set to 0.5 m. Figure 7a shows a profile view of the experimental plot before lodging on August 28. The heights of the maize plants before the lodging were uniform without significant difference. Figure 7b shows a profile view of the experimental plot after the artificial lodging treatment on August 28. Compared with those of the control group, the plant heights of the plot were changed after the lodging treatment, and more severe lodging resulted in lower plant heights. Figure 7c shows the profile of the experimental plot on September 14 after a period of artificial lodging. Because lodging occurred during the tasseling stage, the maize was in the vegetative growth stage at that time, and the maize itself had the ability to recover after lodging. Compared with those immediately after artificial lodging, the plant heights in the lodging area changed significantly. The three profiles (a), (b), and (c) in Figure 7 reflect the time series of maize plant height, and the changes of plant height could be clearly observed using the point cloud data.

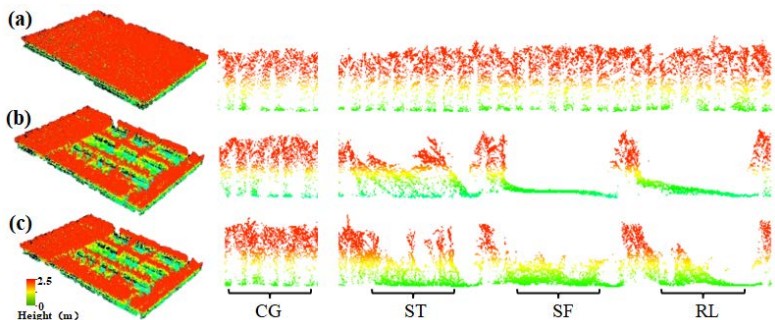

**Figure 7.** Experimental area profile. (**a**) No lodging. (**b**) After artificial lodging. (**c**) Lodging for a period of time. Note: CG: control group; ST: stem tilt; SF: stem folding; RL: root lodging.

### 3.2. Canopy Height Change

The maize canopy height of each flight was obtained from the CHM, and the results are shown in Figure 8. Figure 8a is the maize canopy height without lodging on August 28. When there was no lodging, the maize canopy height was more uniform and the texture features were more regular. The range of maize canopy height estimated by LiDAR is 2.01~2.28 m. Figure 8b shows the maize canopy height after artificial lodging treatment on August 28. Compared with the normal plant heights, the ST, SF, and RL plant heights were significantly reduced. The plant heights of SF and RL were close to zero. The height range of the canopy of lodged maize estimated by LiDAR is: ST 1.21~1.47 m; SF 0.06~0.17 m; RL 0.08~0.18 m. Figure 8c shows the maize canopy heights on September 14. Compared with those shown in Figure 8b, the maize plant heights in each lodging area had recovered to a certain extent. Because of the inconsistent recovery ability of each maize lodging type, the plant heights were different, and the textures were less even. The height range of the canopy of lodged maize estimated by LiDAR is: CG 2.04~2.42 m; ST 1.70~2.18 m; SF 1.05~1.32 m; RL 1.17~1.59 m.

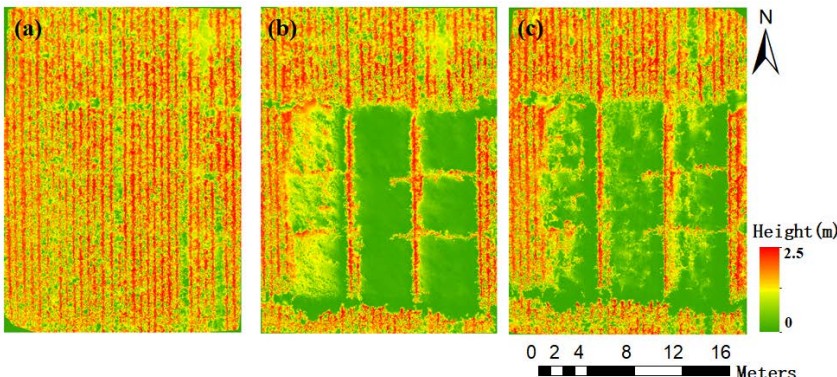

**Figure 8.** Canopy height changes in the experimental plots. (**a**) No lodging. (**b**) After artificial lodging. (**c**) Lodging for a period of time.

### 3.3. Plant Height Verification

The plant height of the maize measured on September 14 and the plant height estimated by UAV-LiDAR were verified. To verify the accuracy of the canopy heights obtained from the UAV-LiDAR data. A total of 50 plant height data were used for verification (including 15 RL, 15 SF, 15 ST, and 5 CG), and the results are shown in Figure 9. There was a significant positive correlation between the measured maize plant heights of lodged maize and the estimated plant height of UAV-LiDAR, and the correlation was 0.98 ($P < 0.01$). The verification results are $R^2 = 0.964$, RMSE = 0.127, and nRMSE = 7.449%, and the plant height estimation accuracy was very high.

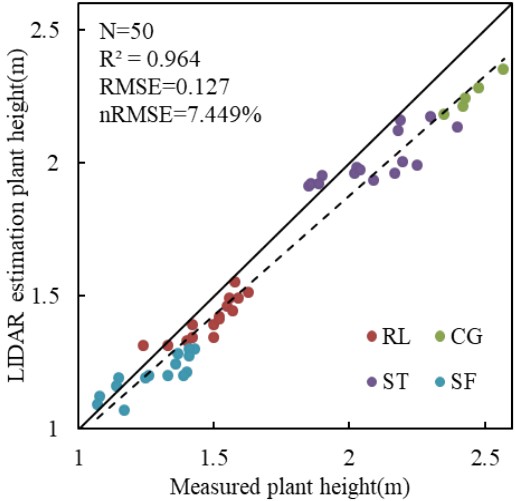

**Figure 9.** Comparison between measured plant height and LiDAR estimation plant height. Note: CG: control group; ST: stem tilt; SF: stem folding; RL: root lodging.

As shown in Figure 9, compared with the measured maize height, the overall plant height estimated by UAV-LiDAR was relatively small. The plant heights of ST were significantly higher than those of SF and RL. Among them, the plant heights of SF were the lowest among the three lodging treatments, indicating that in terms of the restoration capacity of lodging maize plant height, RL > SF. Compared with RL and SF, ST was the least affected by lodging stress, and plant height recovery was not particularly obvious.

## 4. Discussion

UAV remote sensing technology has the advantages of high timeliness, high spatiotemporal resolution, low-altitude flight under clouds, and high maneuverability. The acquired remote sensing

images have advantages such as high resolution, high overlap rate, low cost, and good universality, and can provide additional remote sensing information, such as the fine crop textures, that satellite remote sensing cannot [17,20,52]. It is non-invasive and non-destructive in measurement, and can quickly obtain crop information, which can be well applied to the field of precision agricultural remote sensing monitoring [29,30,36]. Plant height is one of the main types of data describing crop growth status. Being able to obtain crop plant height data quickly and accurately has great significance for monitoring crop growth and fine management of modern agriculture [38,53].

After the lodging of crops, the stems and leaves overlap with each other, which disrupts the normal distribution order of leaves in space, destroys the spatial structure of the plants, and leads to increased population density and ground coverage. UAV digital images (RGB) often use the structure-from-motion method for 3D reconstruction to calculate plant height. When the same point is viewed from two different directions, if blocked, structure from motion will not penetrate into the dense canopy [54–56]. The structure-from-motion technique generally provides a good description of the digital surface model, but accessing the digital terrain model is only possible when the ground is clearly visible [57]. Therefore, there are many limitations in using UAV digital images to monitor the plant height of lodging maize. For example, Esther et al. [58] used UAV RGB images to estimate canopy heights of grasslands in northern Hessen, Germany, and the estimation accuracy of plant height showed $R^2$ = 0.56 and RMSE = 0.13 m. LiDAR is not affected by external light conditions, and provides a method to accurately estimate crop heights [35,36,38]. It can penetrate the crop canopy and reach the ground, providing both a crop surface model and a digital terrain model, and the plant height can then be calculated based on the difference between the crop surface model and the digital terrain model. Thus, the plant height can be measured directly [36]. The accuracy of using LiDAR to measure plant height can reach several centimeters [59,60]. For example, Tilly et al. [39] used terrestrial laser scanner (TLS) technology to monitor plant height in rice fields, and there was a strong correlation between TLS estimated plant height and measured plant height. The results showed that TLS has obvious advantages in measuring crop plant height. Sun et al. [41] developed a cotton high-throughput phenotype (HTP) system based on LiDAR data for estimating cotton plant height, and in terms of the average accuracy of plant height estimation, $R^2$ was 0.98 (RMSE = 65 mm), which indicates accurate plant height measurements.

In many studies, whether RGB images or LiDAR data are used, the estimated plant height is lower than the measured plant height. For example, Niu et al. [61] used UAV RGB images to extract maize plant height, and found that compared with the measured plant heights, the estimated values of plant heights obtained from the UAV RGB images were relatively low. The reason may be that the highest position of maize appeared in the leaf tip during the study period, but because of the limitation of point cloud accuracy, the leaf tip position may have been difficult to detect. Walter et al. [62] found that LiDAR is suitable for measuring relative plant height. As the plant height increases, LiDAR will underestimate the plant height, usually with a decrease of about 10 cm. This decrease is likely caused by the data cleaning process and the lidar crop height (LCH) algorithm function, and although it is not an issue if relative crop height (CH) is desired, if an absolute measurement of CH is required, this discrepancy needs to be accounted for. In this study, there is a similar situation for maize plant height based on CHM extraction. It was found that with increased height of maize plants, the error of the plant height retrieved from LiDAR compared with the actual height increased. There may be two reasons for this increased error. (1) When plant height is measured in the field, there is no lodging maize height higher than the height of the surveyor. When a telescopic leveling ruler is used, the leveling ruler is placed behind the maize plant, and the surveyor looks up at the reading, which leads to a large reading. (2) When the plant height of maize is measured, the highest point selected is the male tassel, and corresponding spatial structure is small. When the CHM was used to invert plant height, information on the spatial structure of the tassel was lost, resulting in low plant height. With more severe the lodging, the spatial structure of the tassel is denser, and the loss of

spatial structure information is smaller. Therefore, compared with that of CG and ST, the plant height estimation accuracy of RL and SF is higher.

After lodging in the tasseling stage, and in the vegetative growth stage, the maize itself has a certain ability to recover. An intuitive manifestation of this ability is that the plant height shows a certain recovery, but this plant height recovery is only reflected in the maize canopy; the rhizome recovery is weak. Therefore, after lodging for a period of time, the height of the maize plants changed significantly only at the canopy position. Consequently, in the canopy height change chart (Figure 8), only a portion of the plant heights changed significantly. Because the plant height recovery of the lodged maize stalks is not obvious and is difficult to distinguish, it will affect the judgment of maize disasters with different lodging degrees. How this can be distinguished will be the focus of our next work. In addition, this study found that the plant height recovery ability of maize with different types of lodging differed. After lodging, the lodging severity of RL was higher than that of SF, and the plant height was lower. However, according to the measured data and LiDAR estimation data after a period of lodging, the plant height of RL was significantly higher than that of SF (Figure 9). The reason is that SF destroyed the transport system of the crops, resulting in blocked transportation of water, nutrients, and other materials, affecting plant growth and leading to lower plant heights [63,64].

## 5. Conclusions

In this study, based on field experiments of artificial lodging maize, multi-stage UAV-LiDAR data were obtained. The plant height changes of different lodging types of maize were analyzed, and the maize canopy height was estimated. The accuracy was verified using ground-based measured data. The results show the following.

(1) Lodged maize has the ability to restore plant height, and canopy recovery is apparent, but the stem and root recovery is weak.

(2) The UAV-LiDAR data can reflect the temporal changes of lodged maize plant height and the plant height restoration ability of different lodging types, and in terms of plant height restoration ability, RL > SF.

(3) The UAV-LiDAR data can provide accurate estimated of lodged maize plant height. The plant height estimated accuracy parameters were $R^2 = 0.964$, RMSE = 0.127, and nRMSE = 7.449%.

In contrast with optical sensors, LiDAR is not limited by the surrounding environmental conditions, has stronger penetrating ability, can obtain the vertical structure information of vegetation, can more accurately extract crop height, can more intuitively reflect lodging, and can provide new methods for maize lodging monitoring. However, because of the high cost of LiDAR and the short UAV flight time, it is currently only applicable to small-scale lodging monitoring. Application in large-scale lodging monitoring is still a difficult problem to solve. In addition, how LiDAR data can be used to extract changes in other structural parameters of lodging maize will be the focus of further research.

**Author Contributions:** Conceptualization, X.G.; Data curation, S.C.; Investigation, L.Z., M.S., and Q.S.; Methodology, L.Z., S.C., and M.Y.S.; Validation, L.Z., M.S., and Q.S.; Writing—original draft, L.Z. and X.G.; Writing—review and editing, G.Y. All authors have read and agreed to the published version of the manuscript.

**Funding:** This study was funded by the National Natural Science Foundation of China (41571323), Beijing Natural Science Foundation (6172011).

**Conflicts of Interest:** The authors declare no conflict of interest.

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
