# Peer review of "Analysis of Plant Height Changes of Lodged Maize Using UAV-LiDAR Data"

_agriculture, doi:10.3390/agriculture10050146_

Round 1

Reviewer 1 Report

Line 52 = In the field-based methods, relevant….

Line 56 = non-lodged plots

Line 85 = Provide a reference

Line 103-104 = After lodging, the morphology of a maize plant can change drastically; from upright to slight leaning to flat horizontal on the ground.

Line 127-128 = to obtain height of lodged maize at a single growth stage

Line 129 = plant height recovery for different lodging degrees

Line 130 = to estimate the height…

Line 150 = each of which were….

Line 162 = …field experience. Remove artificially lodging maize or rephrase. The sentence does not make sense.

Line 164-165 = remove to measure the canopy height of lodged maize. Rephrase. Correct the grammar

Line 165-166 = There were 12 plots with 5 measurement in each? This should results in 60 samples. Why were the remaining 10 samples not considered. Please specify.

Line 168 = Types of lodging…

Section 3.3  = In spite of the previous review, the authors fail to validate the model with an independent set of observations.  Usually, 75% of the total observations need to be used for calibration while 25% should reserved for validation. Since the sample number in this study is small (50), I suggest authors to use cross validation method. Therefore the R2 and RMSE for calibration and validation should be reported i.e. R2Cal, R2CV, RMSECal, RMSECV.

Author Response

Responses and Modifications Based on Reviewers’ Comments

Thank you very much for giving us the chance to revise our manuscript for the publication in Agriculture. And we appreciated the comments and suggestions that raised by the editor on improving our manuscript. Based on the comments and suggestions, we have revised the manuscript, and the modified part is marked in red font in the article.

The black-colored texts are the comments made by editors or reviewer;

The blue-colored texts are the responses and modifications.

Sincerely yours

Longfei Zhou, Xiaohe Gu, Shu Chen, Guijun Yang, Meiyan Shu and Qian Sun

Line 52 = In the field-based methods, relevant….

Line 56 = non-lodged plots

Line 85 = Provide a reference

Line 103-104 = After lodging, the morphology of a maize plant can change drastically; from upright to slight leaning to flat horizontal on the ground.

Line 127-128 = to obtain height of lodged maize at a single growth stage

Line 129 = plant height recovery for different lodging degrees

Line 130 = to estimate the height…

Line 150 = each of which were….

Line 168 = Types of lodging…

Response and modification: Thank you very much for your suggestion. According to your suggestion, we have modified the above questions according to your requirements. For details, please refer to the modification in the article.

 2.Line 162 = …field experience. Remove artificially lodging maize or rephrase. The sentence does not make sense.

Response and modification: Thank you very much for your suggestion. We have modified the above questions according to your requirements. For details, please refer to the modification in the article.

The modified content can be found on lines 161 on page 4.

3.Line 164-165 = remove to measure the canopy height of lodged maize. Rephrase. Correct the grammar

Response and modification: Thank you very much for your suggestion. According to your suggestion, we have modified the above questions according to your requirements. For details, please refer to the modification in the article.

The modified content can be found on lines 162-163 on page 4.

4.Line 165-166 = There were 12 plots with 5 measurement in each? This should results in 60 samples. Why were the remaining 10 samples not considered. Please specify.

Response and modification: Thank you very much for your suggestion. Since the difference in normal maize plant height is not obvious, we regard the normal maize plots as a whole (as shown in Figure 1(a)), and it is regarded as a plot when obtaining plant height data. Therefore, we got a total of 50 samples. We have modified the relevant contents in the article.

The modified content can be found on lines 157-158 on page 4.

5.Section 3.3 = In spite of the previous review, the authors fail to validate the model with an independent set of observations.  Usually, 75% of the total observations need to be used for calibration while 25% should reserved for validation. Since the sample number in this study is small (50), I suggest authors to use cross validation method. Therefore the R2 and RMSE for calibration and validation should be reported i.e. R2Cal, R2CV, RMSECal, RMSECV.

Response and modification: Thank you very much for your suggestion. In this paper, we use the CHM method in LIDAR360 software to obtain the lodged maize canopy height from UAV-LIDAR data. In LIDAR360 software, the plant height was extracted using the software's existing algorithms. The plant height model is not established in this article. The modified content can be found on lines 216 on page 7. We use 50 artificially measured plant heights to verify the estimated plant height of UAV-LIDAR. The purpose is to verify the accuracy of UAV-LIDAR plant height estimation. We refer to the research of Walter and Yuan.

Walter, J.D.C.; James, E.; Mcdonald, G.; Kuchel, H. Estimating biomass and canopy height with LiDAR for field crop breeding. Front. Plant Sci. 2019, 10, 1145.

Yuan, W.A.; Li, J.T.; Bhatta, M.; Shi, Y.Y.; Baenziger, P.S.; Ge, Y.F. Wheat Height Estimation Using LiDAR in Comparison to Ultrasonic Sensor and UAS. Sensors. 2018, 18(11).

Thank you for your suggestion. Your suggestion is very important. In my future work, I will improve the scientific research level and achieve more results according to your suggestions.

Thank you very much for your comments and suggestions.

We appreciate for Editors/Reviewers’ warm work earnestly.

Reviewer 2 Report

The authors have improved the draft of the manuscript from first submission. There were some satisfactory changes that were made. However, there are a few areas especially English grammar and style which still requires some work. I have highlighted those sentences in the attached document. I would suggest authors to review the draft again by keeping the language style and grammar in mind.

Author Response

Responses and Modifications Based on Reviewers’ Comments

Thank you very much for giving us the chance to revise our manuscript for the publication in Agriculture. And we appreciated the comments and suggestions that raised by the editor on improving our manuscript. Based on the comments and suggestions, we have revised the manuscript, and the modified part is marked in red font in the article.

The black-colored texts are the comments made by editors or reviewer;

The blue-colored texts are the responses and modifications.

Sincerely yours

Longfei Zhou, Xiaohe Gu, Shu Chen, Guijun Yang, Meiyan Shu and Qian Sun

1.L51-52: In linr 51 you introduced two methods but in the subsequent sentence a new “artifical method” is introduced. There are two issues here:a) what is the connection b/w the two methods and artifical method. b) what is artifical method?

For point b i really don’t like the loose use of word artificial here. Please clrafy before you proceed with the further uise of this word elsewehere.

Response and modification: Thank you very much for your suggestion. What we mean by "artificial methods" refers to manual field investigations, that is, "field-based methods." We have modified the above questions according to your requirements. For details, please refer to the modification in the article.

The modified content can be found on lines 52 on page 2.

 2.L57: this entrire paragraph is too big to digest properly. Please consider splitting into smaller chunks.

Response and modification: Thank you very much for your suggestion. We have modified the above questions according to your requirements. For details, please refer to the modification in the article.

3.L149: “there were three types of lodging”

This sentence doesn’t make sense. Were there three diffrernt lodging evaluation methods that were tested ? Please change the phrasing to clearly reflect the methodology.

Response and modification: Thank you very much for your suggestion. We did not test using three different lodging evaluation methods. We have modified the above questions according to your requirements. For details, please refer to the modification in the article.

The modified content can be found on lines 147-149 on page 4.

4.L161: The sentence needs rephrasing. The grammar is off.

Response and modification: Thank you very much for your suggestion. We have modified the above questions according to your requirements. For details, please refer to the modification in the article.

The modified content can be found on lines 161 on page 4.

We appreciate for Editors/Reviewers’ warm work earnestly, and hope that the correction will meet with approval.

Once again, thank you very much for your comments and suggestions.

This manuscript is a resubmission of an earlier submission. The following is a list of the peer review reports and author responses from that submission.

Round 1

Reviewer 1 Report

Reviewer #1: This paper is a useful example of the use of UAV-LiDAR data for lodging assessment in maize by quantitative assessment of plant height. The rationale for doing the work is justified, as the remote sensing-based solution for lodging assessment is needed but still missing, and exploring the benefits of active remote sensing data is interesting. The paper is well explained in most parts. A weakness of the manuscript is that the validation of the model developed is missing and the text is too wordy in some places. The introduction needs references more recent references that have worked on remote sensing-based crop lodging assessment (airborne or satellite scale) and the research gap should be emphasized more clearly.

The specific comments are as follows.

Specific comments:

Line 2: The title should be changed from ‘…Lodging Maize…’ to ‘…Lodged Maize…’

Line 16: ‘The purposes of this paper were to’ should be ‘The aim of this study was to’

Line 17: ‘of multi-stage lodging’ should be ‘of lodged maize’

Line 38: ‘extreme weather’ should be ‘extreme weather events’

Line 43: ‘in decreases’ should be ‘in a decrease’

Line 44: ‘affect’ should be ‘affecting’

Line  46: The term light and serious cases should be replaced with mild/moderate and severe/very respectively. Also, the authors should define the terms using the literature. For instance, mild/moderate can be when 30% area is lodged or the crop angle is, say 30 degrees from the vertical (the figures are not real)

Line 50: ‘after’ should ‘in’

Line 51: The usage of the term artificial as a method is not appropriate. Should be either field-based methods or visual inspection

Line 53: ‘Lodging’ here and elsewhere (in a similar context) should be ‘lodged’. For instance, lodged maize, lodged areas, lodged plots, lodged crop, etc. While, in the case like most crop lodging studies, maize lodging, lodging in maize, etc., the usage of lodging is correct.

Line 55: characteristic remote sensing-derived variables.

Line 62: define lodging degrees before

Line 79: The reason why the optical data can not be useful for lodging needs to be reformulated and clearly explained.

Line 81: ‘an error’

Line 88-91: Sentence is too long. Should be split into two and clearly formulated.

Line 91: what is hierarchical monitoring of maize lodging degree?

Line 91-94: This is not entirely true. Recently there have been one or two studies which have investigated crop lodging quantitatively using SAR using Sentinel-1/Radarsat-2 data. The authors should cite at least one.

Line 96: the term lower plant height is ambiguous. The authors should be specific.

Line 95-98: The reduction in plant height due to lodging is well known. However, the authors should justify why the use of plant height and not, for instance, the crop angle of inclination is a better parameter to investigate in terms of maize lodging. Plant height is dependent on crop variety and growth stage while the crop angle is not. Please justify and incorporate it in the introduction.

Line 101: remove ‘for calculation’

Line 117: Instead of multi-period, authors should say point cloud data across different growth stages. Also, state clearly that the quantitative plant height retrieval was limited to a single growth stage and not multiple growth stages.

Line 119: changes in plant height

Line 120: evaluate is a better term than judge

Line 121: Çhange to ‘(2) to use LiDAR point cloud data to retrieve the height of lodged maize’

Line 122: ‘in monitoring the height of lodged maize’

Line 130-132: Does it mean the study was conducted in only one maize field? Please specify, along with the size of the field under study.

Line 131: the 70 and 30cm spacing is not clear

Section 2.2: Explain either graphically or in text, how the height is being measured for each lodging treatment (SL, SF and RL).

Line 140-141: ‘the whole plant falls on the ground’ does not necessarily mean root lodging. The authors can say ‘all the roots were uprooted’ or ‘the entire anchorage system was destroyed’ or something else.

Line 139-143: The definitions of strong to weak are not clear. Is stem lodging weak? Or the whole plant falls to the ground is weak? Also, lodging is primarily of two kinds: root and stem lodging. Stem folding falls within the category of stem lodging as well. The way it has been stated here is misleading. The two kinds of stem lodging here can be referred to as mild and severe stem lodging instead.

Line 145: How many plots were considered in total? Were there any subplots, considering the large variability in the plot size of 5x5m?

Figure 1 caption says much more than just the location of the study area. Put that info in the caption.

Line 155: Why was the plant height data on August 28 not collected? What were the growth stages at these two dates?

Line 156: ‘the plant height of maize’

Line 157: It is better to say that the plant height was measured at five sampling locations selected randomly in each plot. Were these values treated as individual plant height values or were averaged for each plot. In either case, state how many plant height readings were recorded/available for the final regression analysis.

Line 159-160: The definition of the distance is not clear. Please re-phrase. What is the route distance?

Table 1 The spot diameter and average ground distance should be removed from the table and can be mentioned in the text.

Line 195: Rather than saying the CHM methods, authors should say that a CHM was constructed

Line 211-214: The sentence needs to be fragmented. Also, the dependent and independent variables for the regression analysis are not clear. Please rephrase. It’s also important to justify if the sample size used for the model development and model validation was statistically significant

Line 226: represents

Line 242-243: reflects the time series of maize plant height.

Section 3.2. The authors should specify the observed range of crop height in the three scenarios presented (although it is not clear if the authors measured plant height in the first 2 scenarios). Line 263 and elsewhere: plant height inverted from LiDAR data.

Line 267: accuracy was very high.

Line 283: Please explain how crop height can explain the crop growth status?

Line 297: Remove cm from the R2 value

Section 3.3 The results are lacking the validation of the regression model. The authors should validate the model with sufficient data points and make it clear in the text.

The manuscript should be checked by a mother tongue speaker.

In summary, the manuscript needs serious revision to proceed.

Reviewer 2 Report

The topic of this study is highly relevant to the high-throughput phenotyping  and agriculture community. Maize lodging is a serious problem in terms of  difficulty in estimation within the field conditions. As such the topic of this study is important.

Overall this paper is nicely structured. I am highlighting here a few shortcomings; for detailed comments please refer to the pdf document.

There are some methodological gaps/details that need some explanation from authors. There are many places where critical methodological details are missing to the extent that it is harder to make proper conclusions. In addition there are minor grammatical/style issues which the authors can improve upon.
